# Conversation Understanding using Relational Temporal Graph Neural Networks with Auxiliary Cross-Modality Interaction

**Cam-Van Thi Nguyen**[1] , **Anh-Tuan Mai**[1,2], **The-Son Le**[1]
**Hai-Dang Kieu**[1], **Duc-Trong Le**[1]

[1]VNU University of Engineering and Technology, Hanoi, Vietnam
[2]FPT Software AI Center
{vanntc, 20020269, 21020089, dangkh_uet, trongld}@vnu.edu.vn

## Abstract

Emotion recognition is a crucial task for human conversation understanding. It becomes more challenging with the notion of multimodal data, e.g., language, voice, and facial expressions. As a typical solution, the global- and the local context information are exploited to predict the emotional label for every single sentence, i.e., utterance, in the dialogue. Specifically, the global representation could be captured via modeling of cross-modal interactions at the conversation level. The local one is often inferred using the temporal information of speakers or emotional shifts, which neglects vital factors at the utterance level. Additionally, most existing approaches take fused features of multiple modalities in an unified input without leveraging modality-specific representations. Motivating from these problems, we propose the Relational Temporal Graph Neural Network with Auxiliary Cross-Modality Interaction (CORECT), an novel neural network framework that effectively captures conversation-level cross-modality interactions and utterance-level temporal dependencies with the modality-specific manner for conversation understanding. Extensive experiments demonstrate the effectiveness of CORECT via its state-of-the-art results on the IEMOCAP and CMU-MOSEI datasets for the multimodal ERC task.

## 1 Introduction

Our social interactions and relationships are all influenced by emotions. Given the transcript of a conversation and speaker information for each constituent utterance, the task of Emotion Recognition in Conversations (ERC) aims to identify the emotion expressed in each utterance from a predefined set of emotions (Poria et al., 2019). The multimodal nature of human communication, which involves verbal/textual, facial expressions, vocal/acoustic, bodily/postural, and symbolic/pictorial expressions, adds complexity to the task of Emotion Recognition in Conversations (ERC) (Wang et al., 2022).

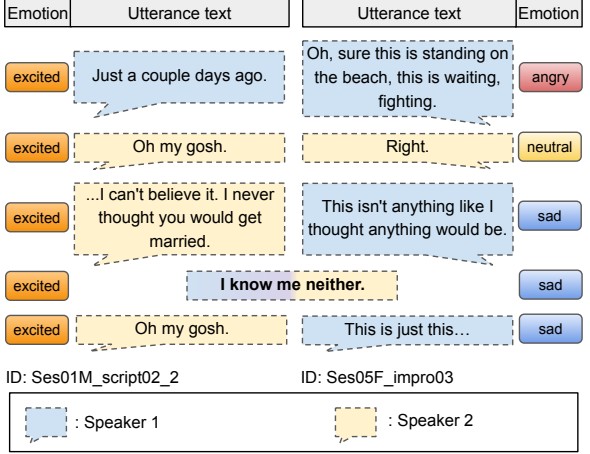

Figure 1: Examples of temporal effects on conversations

Multimodal ERC, which aims to automatically detect the a speaker's emotional state during a conversation using information from text content, facial expressions, and audio signals, has garnered significant attention and research in recent years and has been applied to many real-world scenarios (Sharma and Dhall, 2021; Joshi et al., 2022).

Massive methods have been developed to model conversation's context. These approaches can be categorized into two main groups: graph-based methods (Ghosal et al., 2019; Zhang et al., 2019; Shen et al., 2021b) and recurrence-based methods (Hazarika et al., 2018a; Ghosal et al., 2020; Majumder et al., 2019; Hu et al., 2021). In addition, there have been advancements in multimodal models that leverage the dependencies and complementarities of multiple modalities to improve the ERC performance (Poria et al., 2017; Hazarika et al., 2018b; Zadeh et al., 2018). One limitation of these methods is their heavy reliance on nearby utterances when updating the state of the query utterance, which can restrict their overall performance. Recently, Graph Neural Network (GNN)-based methods have been proposed for the multimodal ERC task due to their ability to capture

long-distance contextual information through their relational modeling capabilities. However, those models rely on fused inputs being treated as a single node in the graph (Ghosal et al., 2019; Joshi et al., 2022), which limits their ability to capture modality-specific representations and ultimately hampers their overall performance.

The temporal aspect of conversations is crucial, as past and future utterances can significantly influence the query utterance as Figure 1. The sentence "*I know me neither*" appears with opposing labels on different dialogues, which could be caused by sequential effects from previous or future steps. There are only a few methods that take into account the temporal aspect of conversations. MMGCN (Wei et al., 2019) represents modality-specific features as graph nodes but overlooks the temporal factor. DAG-ERC (Shen et al., 2021b) incorporates temporal information, but focuses solely on text modality. Recently, COGMEN (Joshi et al., 2022) proposes to learn contextual, inter-speaker, and intra-speaker relations, but neglects modality-specific features and partially utilizes cross-modal information by fusing all modalities' representations at the input stage.

The aforementioned limitations motivate us to propose a **CO**nversation understanding model using **RE**lational **T**emporal Graph Neural Network with Auxiliary **C**ross-Modality Interaction (**CORECT**). It comprises two key components: the *(i) Relational Temporal Graph Convolutional Network (RT-GCN)*; and the *(ii) Pairwise Cross-modal Feature Interaction (P-CM)*. The *RT-GCN* module is based on RGCNs (Schlichtkrull et al., 2018) and GraphTransformer (Yun et al., 2019) while the *P-CM* is built upon (Tsai et al., 2019). Overall, our main contributions are as follows:

- We propose the CORECT framework for Multimodal ERC, which concurrently exploit the utterance-level local context feature from multimodal interactions with temporal dependencies via RT-GCN, and the cross-modal global context feature at the conversation level by P-CM. These features are aggregated to enhance the performance of the utterance-level emotional recognition.

- We conduct extensive experiments to show that CORECT consistently outperforms the previous SOTA baselines on the two publicly real-life datasets, including IEMOCAP and CMU-MOSEI, for the multimodal ERC task.

- We conduct ablation studies to investigate the effect of various components and modalities on CORECT for conversation understanding.

## 2 Related Works

This section presents a literature review on Multimodal Emotion Recognition (ERC) and the application of Graph Neural Networks for ERC.

### 2.1 Multimodal Emotion Recognition in Conversation

The complexity of conversations, with multiple speakers, dynamic interactions, and contextual dependencies, presents challenges for the ERC task. There are efforts to model the conversation context in ERC, with a primary focus on the textual modality. Several notable approaches include CMN (Hazarika et al., 2018b), DialogueGCN (Ghosal et al., 2019), COSMIC (Ghosal et al., 2020), DialogueXL (Shen et al., 2021a), DialogueCRN (Hu et al., 2021), DAG-ERC (Shen et al., 2021b).

Multimodal machine learning has gained popularity due to its ability to address the limitations of unimodal approaches in capturing complex real-world phenomena (Baltrušaitis et al., 2018). It is recognized that human perception and understanding are influenced by the integration of multiple sensory inputs. There have been several notable approaches that aim to harness the power of multiple modalities in various applications (Poria et al., 2017; Zadeh et al., 2018; Majumder et al., 2019), etc. CMN (Hazarika et al., 2018b) combines features from different modalities by concatenating them directly and utilizes the Gated Recurrent Unit (GRU) to model contextual information. ICON (Hazarika et al., 2018a) extracts multimodal conversation features and employs global memories to model emotional influences hierarchically, resulting in improved performance for utterance-video emotion recognition. ConGCN (Zhang et al., 2019) models utterances and speakers as nodes in a graph, capturing context dependencies and speaker dependencies as edges. However, ConGCN focuses only on textual and acoustic features and does not consider other modalities. MMGCN (Wei et al., 2019), on the other hand, is a graph convolutional network (GCN)-based model that effectively captures both long-distance contextual information and multimodal interactive information.

More recently, Lian et al. Lian et al. (2022) propose a novel framework that combines semi-

supervised learning with multimodal interactions. However, it currently addresses only two modalities, i.e., text and audio, with visual information reserved for future work. Shi and Huang (2023) introduces MultiEMO, an attention-based multimodal fusion framework that effectively integrates information from textual, audio and visual modalities. However, neither of these models addresses the temporal aspect in conversations.

## 2.2 Graph Neural Networks

In the past few years, there has been a growing interest in representing non-Euclidean data as graphs. However, the complexity of graph data has presented challenges for traditional neural network models. From initial research on graph neural networks (GNNs)(Gori et al., 2005; Scarselli et al., 2008), generalizing the operations of deep neural networks were paid attention, such as convolution (Kipf and Welling, 2017), recurrence (Nicolicioiu et al., 2019), and attention (Velickovic et al., 2018), to graph structures. When faced with intricate interdependencies between modalities, GNN is a more efficient approach to exploit the potential of multimodal datasets. The strength of GNNs lies in its ability to capture and model intra-modal and inter-modal interactions. This flexibility makes them an appealing choice for multimodal learning tasks.

There have been extensive studies using the capability of GNNs to model the conversations. DialogueGCN (Ghosal et al., 2019) models conversation using a directed graph with utterances as nodes and dependencies as edges, fitting it into a GCN structure. MMGCN (Wei et al., 2019) adopts an undirected graph to effectively fuse multimodal information and capture long-distance contextual and inter-modal interactions. Lian et al. (2020) proposed a GNN-based architecture for ERC that utilizes both text and speech modalities. Dialogue-CRN (Hu et al., 2021) incorporates multiturn reasoning modules to extract and integrate emotional clues, enabling a comprehensive understanding of the conversational context from a cognitive perspective. MTAG (Yang et al., 2021) is capable of both fusion and alignment of asynchronously distributed multimodal sequential data. COGMEN (Joshi et al., 2022) uses GNN-based architecture to model complex dependencies, including local and global information in a conversation. Chen et al. (2023) presents Multivariate Multi-frequency Multimodal Graph Neural Network, M$^3$Net for short,

to explore the relationships between modalities and context. However, it primarily focuses on modality-level interactions and does not consider the temporal aspect within the graph.

## 3 Methodology

Figure 2 illustrates the architecture of CORECT to tackle the multimodal ERC task. It consists of main components namely Relational Temporal Graph Convolution Network (RT-GCN) and Pairwise Cross-modal Feature Interaction. For a given utterance in a dialogue, the former is to learn the local-context representation via leveraging various topological relations between utterances and modalities, while the latter infers the cross-modal global-context representation from the whole dialogue.

Given a multi-speaker conversation $C$ consisting of $N$ utterances $[u_1, u_2, \ldots, u_N]$, let us denote $S$ as the respective set of speakers. Each utterance $u_i$ is associated with three modalities, including audio $(a)$, visual $(v)$, and textual $(l)$, that can be represented as $u_i^a, u_i^v, u_i^l$ respectively. Using local- and global context representations, the ERC task aims to predict the label for every utterance $u_i \in C$ from a set of $M$ predefined emotional labels $Y = [y_1, y_2, \ldots, y_M]$.

### 3.1 Utterance-level Feature Extraction

Here, we perform pre-processing procedures to extract utterance-level features to facilitate the learning of CORECT in the next section.

#### 3.1.1 Unimodal Encoder

Given an utterance $u_i$, each data modality manifests a view of its nature. To capture this value, we employ dedicated unimodal encoders, which generate utterance-level features, namely $x_i^a \in \mathbb{R}^{d_a}$, $x_i^v \in \mathbb{R}^{d_v}$, $x_i^l \in \mathbb{R}^{d_l}$ for the acoustic, visual, and lexical modalities respectively, and $d_a, d_v, d_l$ are the dimensions of the extracted features for each modality.

For textual modality, we utilize a Transformer (Vaswani et al., 2017) as the unimodal encoder to extract the semantic feature $x_i^l$ from $u_i^l$ as follows:

$$x_i^l = \textbf{Transformer}(u_i^l, \mathbf{W}_{trans}^l) \qquad (1)$$

where $\mathbf{W}_{trans}^l$ is the parameter of Transformer to be learned.

For acoustic and visual modalities, we employ a fully-connected network as the unimodal encoder

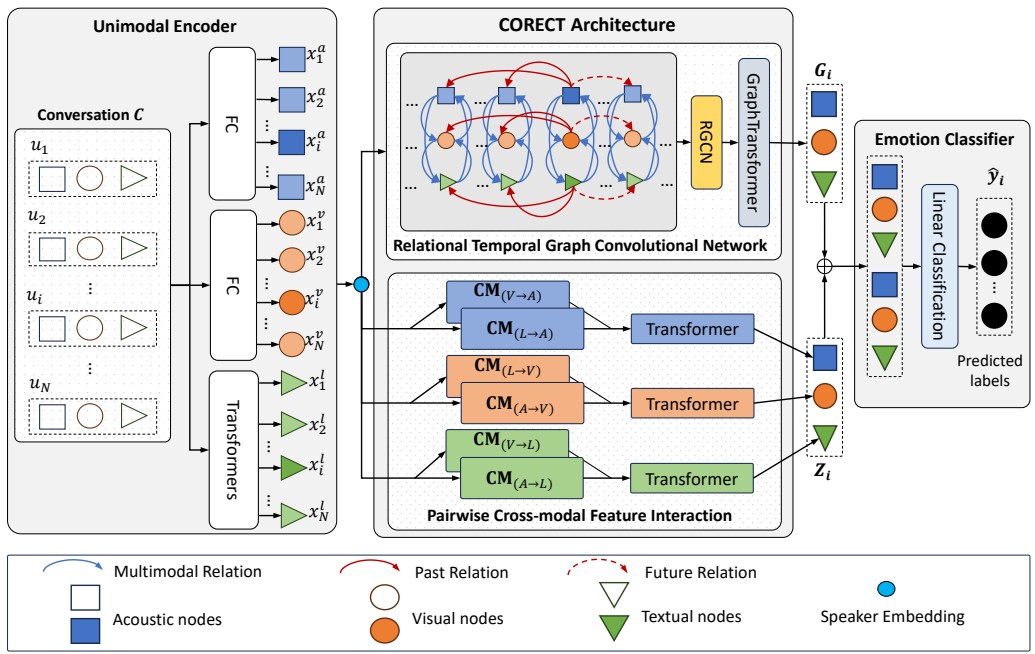

Figure 2: Framework illustration of CORECT for the multimodal emotion recognition in conversations

to extract context features for each modality type via the following procedure:

$$x_i^\tau = \mathbf{FC}(u_i^\tau; \mathbf{W}_{fc}^\tau), \tau \in \{a, v\} \qquad (2)$$

where $\mathbf{FC}$ is the fully connected network, $\mathbf{W}_{fc}^\tau \in \mathbb{R}^{d_\tau \times d_{in}^\tau}$ are trainable parameters; $d_{in}^\tau$ is the input dimension of modality $\tau$

### 3.1.2 Speaker Embedding

Inspired by MMGCN (Wei et al., 2019), we leverage the significance of speaker information. Let us define **Embedding** as a procedure that takes the identity of speakers and produce the respective latent representations. The embedding of multi-speaker could be inferred as:

$$\mathcal{S}_{emb} = \mathbf{Embedding}(S, \mathcal{N}_S) \qquad (3)$$

where $\mathcal{S}_{emb} \in \mathbb{R}^{N \times \mathcal{N}_S}$ and $\mathcal{N}_S$ is the total number of participants in the conversation. The extracted utterance-level feature could be enhanced by adding the corresponding speaker embedding:

$$\mathbf{X}_\tau = \eta \mathcal{S}_{emb} + \mathcal{X}_\tau, \tau \in \{a, v, l\} \qquad (4)$$

where $\mathcal{X}_\tau \in \mathbb{R}^{N \times d_\tau}$ refers to the global-context representation from the whole dialogue obtained from the respective unimodal encoder; $\mathbf{X}_\tau$ represents the enhanced representation with the inclusion of the speaker embedding; $\eta \in [0, 1]$ indicates the contribution ratio.

### 3.2 Relational Temporal Graph Convolutional Network (RT-GCN)

RT-GCN is proposed to capture local context information for each utterance in the conversation via exploiting the multimodal graph between utterances and their modalities.

#### 3.2.1 Multimodal Graph Construction

Let us denote $\mathcal{G}(\mathcal{V}, \mathcal{R}, \mathcal{E})$ as the multimodal graph built from conversations, where $\{\mathcal{V}, \mathcal{E}, \mathcal{R}\}$ refers to the set of utterance nodes with the three modality types ($|\mathcal{V}| = 3 \times N$), the set of edges and their relation types. Figure 3 provides an illustrative example of the relations represented on the constructed graph.

**Nodes.** Each utterance $u_i$ generates three nodes $u_i^a$, $u_i^v$, and $u_i^l$, which $x_i^a$, $x_i^v$, and $x_i^l$ are the respective audio, visual, and lexical feature vectors.

**Edges.** The edge $(u_i^\tau, u_j^\tau, r_{ij}) \in \mathcal{E}, \tau \in \{a, v, l\}$ represents the interaction between $u_i^\tau$ and $u_j^\tau$ with the relation type $r_{ij} \in \mathcal{R}$. In the scope of paper, we consider two groups of relations: $\mathcal{R}_{multi}$ and $\mathcal{R}_{temp}$. Specifically, $\mathcal{R}_{multi}$ represents the intra connections between the three modalities within the same utterance, reflecting multimodal interactions. On the other hand, $\mathcal{R}_{temp}$ captures the inter connections between utterances of the same modality within a specified time window. This temporal relationship includes past/previous utterances de-

noted as $\mathcal{P}$ and next/future utterances denoted as $\mathcal{F}$. As a result, there are 15 edge types created with the definitions of the two groups.

*Multimodal Relation.* Emotions in dialogues cannot be solely conveyed through lexical, acoustic, or visual modalities in isolation. The interactions between utterances across different modalities play a crucial role. For example, given an utterance in a graph, its visual node has different interactive magnitude with acoustic- and textual nodes. Additionally, each node has a self-aware connection to reinforce its own information. Therefore, we can formalize 9 edge types of $\mathcal{R}_{multi}$ to capture the multimodal interactions within the dialogue as:

$$\mathcal{R}_{multi} = \begin{cases} \{(u_i^a, u_i^v), (u_i^v, u_i^a), (u_i^a, u_i^a)\} \\ \{(u_i^v, u_i^l), (u_i^l, u_i^v), (u_i^v, u_i^v)\} \\ \{(u_i^l, u_i^a), (u_i^a, u_i^l), (u_i^l, u_i^l)\} \end{cases} \quad (5)$$

*Temporal Relation.* It is vital to have distinct treatment for interactions between nodes that occur in different temporal orders (Poria et al., 2017). To capture this temporal aspect, we set a window slide $[\mathcal{P}, \mathcal{F}]$ to control the number of past/previous and next/future utterances that are set has connection to current node $u_i^\tau$. This window enables us to define the temporal context for each node and capture the relevant information from the dynamic surrounding utterances. Therefore, we have 6 edge types of $\mathcal{R}_{temp}$ as follows:

$$\mathcal{R}_{temp} = \begin{cases} \{(u_j \overset{past}{\to} u_i)^\tau | i - \mathcal{P} < j < i\} \\ \{(u_i \overset{future}{\leftarrow} u_j)^\tau | i < j < i + \mathcal{F}\} \end{cases} \quad (6)$$

where $\tau \in \{a, v, l\}$; $i, j \in \overline{1, N}$; $\overset{future}{\leftarrow}$ and $\overset{past}{\to}$ indicate the past and future relation respectively.

### 3.2.2 Graph Learning

With the objective of leveraging the nuances and variations of heterogeneous interactions between utterances and modalities in the multimodal graph, we seek to employ Relational Graph Convolutional Networks (RGCN) (Schlichtkrull et al., 2018). For each relation type $r \in \mathcal{R}$, node representation is inferred via a mapping function $f(\mathbf{H}, \mathbf{W}_r)$, where $\mathbf{W}_r$ is the weighted matrix. Aggregating all 15 edge types, the final node representation could be computed by $\sum_r^\mathcal{R} f(\mathbf{H}, \mathbf{W}_r)$.

To be more specific, the representation for the $i$-th utterance is inferred as follows:

$$g_i^\tau = \sum_{r \in \mathcal{R}} \sum_{j \in \mathcal{N}_r(i)} \frac{1}{|\mathcal{N}_r(i)|} \mathbf{W}_r \cdot x_i^\tau + \mathbf{W}_0 \cdot x_i^\tau \quad (7)$$

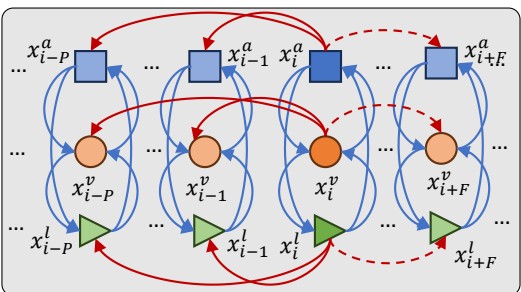

Figure 3: An example construction of a graph illustrating the relationship among utterance nodes representing audio (square), visual (circle), and text (triangle) modalities with window size $[\mathcal{P}, \mathcal{F}] = [2, 1]$ for query utterance $i$-th. The solid blue, solid red, and dashed red arrows indicate cross-modal-, past temporal- and future temporal connections respectively.

where $\mathcal{N}_r(i)$ is the set of the node $i$'s neighbors with the relation $r \in \mathcal{R}$, $\mathbf{W}_0, \mathbf{W}_r \in \mathbb{R}^{d_{h1} \times d_\tau}$ are learnable parameters ($h1$ is the dimension of the hidden layer used by R-GCN), and $x_i^\tau \in \mathbb{R}^{d_\tau \times 1}$ denotes the feature vector of node $u_i^\tau$; $\tau \in \{a, v, l\}$.

To extract rich representations from node features, we utilize a Graph Transformer model (Yun et al., 2019), where each layer comprises a self-attention mechanism followed by feed-forward neural networks. The self-attention mechanism allows vertices to exploit information from neighborhoods as well as capturing local and global patterns in the graph. Given $g_i^\tau$ is the representation of $i^{th}$ utterance with modality $\tau \in \{a, v, l\}$ obtained from RGCNs, its representation is transformed into:

$$o_i^\tau = \|_{c=1}^C [\mathbf{W}_1 g_i^\tau + \sum_{j \in \mathcal{N}(i)} \alpha_{i,j}^\tau \mathbf{W}_2 g_j^\tau] \quad (8)$$

where $\mathbf{W}_1, \mathbf{W}_2 \in \mathbb{R}^{d_{h_2} \times d_{h_1}}$ are learned parameters ($h2$ is the dimension of the hidden layer used by Graph Transformer); $\mathcal{N}(i)$ is the set of nodes that has connections to node $i$; $\|$ is the *concatenation* for $C$ head attention; and the attention coefficient of node $j$, i.e., $\alpha_{i,j}^\tau$, is calculated by the *softmax* activation function:

$$\alpha_{i,j}^\tau = softmax(\frac{(\mathbf{W}_3 g_i^\tau)^\top (\mathbf{W}_4 g_i^\tau)}{\sqrt{d}}) \quad (9)$$

$\mathbf{W}_3, \mathbf{W}_4 \in \mathbb{R}^{d_\alpha \times d_{h_1}}$ are learned parameters.

After the aggregation throughout the whole graph, we obtain new representation vectors:

$$\mathbf{G}^\tau = \{o_1^\tau, o_2^\tau, \dots, o_N^\tau\} \quad (10)$$

where $\tau \in \{a, v, l\}$ indicates the corresponding audio, visual, or textual modality.

## 3.3 Pairwise Cross-modal Feature Interaction

The cross-modal heterogeneities often elevate the difficulty of analyzing human language. Exploiting cross-modality interactions may help to reveal the "unaligned" nature and long-term dependencies across modalities. Inspired by the idea (Tsai et al., 2019), we design the *Pairwise Cross-modal Feature Interaction (P-CM)* method into our proposed framework for conversation understanding. A more detailed illustration of the *P-CM* module is presented in Appendix A.1.2

Given two modalities, e.g., audio $a$ and textual $l$, let us denote $\mathbf{X}^a \in \mathbb{R}^{N \times d_a}$, $\mathbf{X}^l \in \mathbb{R}^{N \times d_l}$ as the respective modality-sensitive representations of the whole conversation using unimodal encoders. Based on the transformer architecture (Vaswani et al., 2017), we define the Queries as $Q^a = \mathbf{X}^a W_{Q^a}$, Keys as $K^l = \mathbf{X}^l W_{K^l}$, and Values as $V^l = \mathbf{X}^l W_{V^l}$. The enriched representation of $\mathbf{X}^a$ once performing cross-modal attention on the modality $a$ by the modality $l$, referred to as $\mathbf{CM}^{l \to a} \in \mathbb{R}^{N \times d_V}$, is computed as:

$$\mathbf{CM}^{l \to a} = \sigma \left( \frac{\mathbf{X}^a \mathbf{W}_{Q^a} (\mathbf{W}_{K^l})^\top (\mathbf{X}^l)^\top}{\sqrt{d_k}} \right) \mathbf{X}_l \mathbf{W}_{V^l} \quad (11)$$

where $\sigma$ is the *softmax* function; $\mathbf{W}_{Q^a} \in \mathbb{R}^{d_a \times d_K}$, $\mathbf{W}_{K^l} \in \mathbb{R}^{d_l \times d_K}$, and $\mathbf{W}_{V^l} \in \mathbb{R}^{d_l \times d_V}$ are learned parameters. The value of $d_Q, d_K, d_V$ is the dimension of queues, keys and values respectively. $\sqrt{d_k}$ is a scaling factor and $d_{(.)}$ is the feature dimension.

To model the cross-modal interactions on unaligned multimodal sequences, e.g., audio, visual, and lexical, we utilize $D$ cross-modal transformer layers. Suppose that $\mathbf{Z}_{[i]}^a$ is the modality-sensitive global-context representation of the whole conversation for the modality $l$ at the $i-$th layer; $\mathbf{Z}_{[0]}^a = \mathbf{X}^a$. The enriched representation of $\mathbf{Z}_l^{[i]}$ denoted as $\mathbf{Z}_{l \to a}^{[i]}$ by applying cross-modal attention of the modality $l$ on the modality $a$ is computed as the following procedure:

$$\mathbf{Z}_{[0]}^{l \to a} = \mathbf{Z}_{[0]}^a$$
$$\overline{\mathbf{Z}}_{[i]}^{l \to a} = \mathbf{CM}_{[i]}^{l \to a}(LN(\mathbf{Z}_{[i-1]}^{l \to a}), LN(\mathbf{Z}_0^{l \to a}))$$
$$\quad + LN(\mathbf{Z}_{[i-1]}^{l \to a})$$
$$\mathbf{Z}_{[i]}^{l \to a} = (LN(\overline{\mathbf{Z}}_{[i]}^{l \to a}))^{FFN} + LN(\overline{\mathbf{Z}}_{[i]}^{l \to a}) \quad (12)$$

where $\overline{\mathbf{Z}}$ is the intermediate representation; $LN$ is a layer normalization (Ba et al., 2016), which helps

| Dataset | Dialogues | | | Utterances | | |
|---|---|---|---|---|---|---|
| | train | valid | test | train | valid | test |
| IEMOCAP (6-way) | 108 | 12 | 31 | 5,146 | 664 | 1,623 |
| IEMOCAP (4-way) | 108 | 12 | 31 | 3,200 | 400 | 943 |
| MOSEI | 2,249 | 300 | 646 | 16,327 | 1,871 | 4,662 |

Table 1: Statistics for IEMOCAP, MOSEI datasets

to stabilize the learning process and enhance the convergence of the model. $LN(\overline{\mathbf{Z}}_{[i]}^{a \to v}))^{FFN}$ expresses the transformation by the position-wise feed-forward block as:

$$LN(\overline{\mathbf{Z}}_{[i]}^{l \to a}))^{FFN} = \max(0, LN(\overline{\mathbf{Z}}_{[i]}^{l \to a}))\mathbf{\Omega}_1$$
$$+ \mathbf{b}_1)\mathbf{\Omega}_2 + \mathbf{b}_2 \quad (13)$$

where $\mathbf{\Omega}_1$ and $\mathbf{\Omega}_2$ are linear projection matrices; $\mathbf{b}_1$ and $\mathbf{b}_2$ are biases.

Likewise, we can easily compute the cross-modal representation $\mathbf{Z}_{a \to l}^{[i]}$, indicating that information from the modality $a$ is transferred to the modality $l$. Finally, we concatenate all representations at the last layer, i.e., the $D-$th layer, to get the final cross-modal global-context representation $\mathbf{Z}_{a \rightleftarrows l}^{[D]}$. For other modality pairs, $\mathbf{Z}_{v \rightleftarrows l}^{[D]}$ and $\mathbf{Z}_{v \rightleftarrows a}^{[D]}$ could be obtained by the similar process.

## 3.4 Multimodal Emotion Classification

The local- and global context representation resulted in by the RT-GCN and P-CM modules are fused together to create the final representation of the conversation:

$$\mathbf{H} = Fusion([\mathbf{G}, \mathbf{Z}]) \quad (14)$$
$$= [\{o_1^\tau, o_2^\tau, \ldots, o_N^\tau\}, \{\mathbf{Z}_{a \rightleftarrows v}^{[D]}, \mathbf{Z}_{v \rightleftarrows l}^{[D]}, \mathbf{Z}_{l \rightleftarrows a}^{[D]}\}]$$

where $\tau \in \{a, v, l\}$; $Fusion$ represents the concatenation method. $\mathbf{H}$ is then fed to a fully connected layer to predict the emotion label $y^i$ for the utterance $u_i$:

$$v_i = ReLU(\mathbf{\Phi}_0 h_i + b_0) \quad (15)$$
$$p_i = \text{softmax}(\mathbf{\Phi}_1 v_i + b_1) \quad (16)$$
$$\hat{y}^i = \text{argmax}(p_i) \quad (17)$$

where $\mathbf{\Phi}_0, \mathbf{\Phi}_1$ are learned parameters.

## 4 Experiments

This section investigate the efficacy of CORECT for the ERC task through extensive experiments in comparing with state-of-the-art (SOTA) baselines.

| Methods | IEMOCAP (6-way) | | | | | | | |
| --- | Happy | Sad | Neutral | Angry | Excited | Frustrated | Acc. (%) | w-F1 (%) |
| --- | --- | --- | --- | --- | --- | --- | --- | --- |
| bc-LSTM (Poria et al., 2017) | 32.63 | 70.34 | 51.14 | 63.44 | 67.91 | 61.06 | 59.58 | 59.10 |
| CMN (Hazarika et al., 2018b) | 30.38 | 62.41 | 52.39 | 59.83 | 60.25 | 60.69 | 56.56 | 56.13 |
| ICON (Hazarika et al., 2018a) | 29.91 | 64.57 | 57.38 | 63.04 | 63.42 | 60.81 | 59.09 | 58.54 |
| DialogueRNN (Majumder et al., 2019) | 33.18 | 78.80 | 59.21 | 65.28 | 71.86 | 58.91 | 63.40 | 62.75 |
| DialogueGCN (Ghosal et al., 2019) | 47.10 | **80.88** | 58.71 | 66.08 | 70.97 | 61.21 | 65.54 | 65.04 |
| MMGCN (Wei et al., 2019) | 45.45 | 77.53 | 61.99 | 66.70 | 72.04 | 64.12 | 65.56 | 65.71 |
| DialogueCRN (Hu et al., 2021) | 51.59 | 74.54 | 62.38 | 67.25 | 73.96 | 59.97 | 65.31 | 65.34 |
| COGMEN (Joshi et al., 2022) | 55.76 | 80.17 | 63.21 | 61.69 | **74.91** | 63.90 | 67.04 | 67.27 |
| **CORECT (Ours)** | **59.30** | 80.53 | **66.94** | **69.59** | 72.69 | **68.50** | **69.93** (↑ 2.89) | **70.02** (↑ 2.75) |

Table 2: The results on IEMOCAP (6-way) multimodal (A+V+T) setting. The results in **bold** indicate the highest performance, while the underlined results represent the second highest performance. The ↑ illustrates the improvement compared to the previous state-of-the-art model.

## 4.1 Experimental Setup

**Dataset.** We investigate two public real-life datasets for the multimodal ERC task including IEMOCAP (Busso et al., 2008) and CMU-MOSEI (Bagher Zadeh et al., 2018). The dataset statistics are given in Table 1.

**IEMOCAP** contains 12 hours of videos of two-way conversations from 10 speakers. Each dialogue is divided into utterances. There are in total 7433 utterances and 151 dialogues. The 6-way dataset contains six emotion labels, i.e., *happy, sad, neutral, angry, excited*, and *frustrated*, assigned to the utterances. As a simplified version, ambiguous pairs such as *(happy, exited)* and *(sad, frustrated)* are merged to form the 4-way dataset.

**CMU-MOSEI** provides annotations for 7 sentiments ranging from highly negative (-3) to highly positive (+3), and 6 emotion labels including *happiness, sadness, disgust, fear, surprise*, and *anger*.

**Evaluation Metrics.** We use *weighted F1-score* (w-F1) and *Accuracy* (Acc.) as evaluation metrics. The w-F1 is computed $\sum_{k=1}^{K} freq_k \times F1_k$, where $freq_k$ is the relative frequency of class $k$. The accuracy is defined as the percentage of correct predictions in the test set.

**Baseline Models.** CORECT is compared against SOTA baselines specific to each dataset. For IEMO-CAP, we consider two model groups namely: i) *RNN-based models* include bc-LSTM (Poria et al., 2017), CMN (Hazarika et al., 2018b), ICON (Hazarika et al., 2018a), DialogueRNN (Majumder et al., 2019); ii) *Graph-based methods* are DialogueGCN (Ghosal et al., 2019), MMGCN (Wei et al., 2019), DialougueCRN (Hu et al., 2021), CHFusion (Majumder et al., 2018), and COGMEN (Joshi et al., 2022). For CMU-MOSEI, we inves-

| Modality Settings | IEMOCAP (4-way) | |
| --- | Acc. (%) | w-F1 (%) |
| --- | --- | --- |
| bc-LSTM (Poria et al., 2017) | 75.20 | 75.13 |
| CHFusion (Majumder et al., 2018) | 76.59 | 76.80 |
| COGMEN (Joshi et al., 2022) | 82.29 | 82.15 |
| **CORECT (Ours)** | **84.73** (↑ 2.44) | **84.64** (↑ 2.49) |

Table 3: The results on the IEMOCAP (4-way) dataset in the multimodal (A+V+T) setting. The ↑ indicates the improvement compared to the previous SOTA model.

tigate multimodal models including Multilogue-Net (Shenoy and Sardana, 2020), TBJE (Delbrouck et al., 2020), and COGMEN (Joshi et al., 2022).

**Implementation Details.** Due to the space limit, the implementation details for feature extraction and interaction are described in Appendix A.1.

## 4.2 Comparison With Baselines

We further qualitatively analyze CORECT and the baselines on the IEMOCAP (4-way), IEMOCAP (6-way) and MOSEI datasets.

**IEMOCAP:** In the case of IEMOCAP (6-way) dataset (Table 2), CORECT performs better than the previous baselines in terms of F1 score for individual labels, excepts the *Sad* and the *Excited* labels. The reason could be the ambiguity between similar emotions, such as *Happy* & *Excited*, as well as *Sad* & *Frustrated* (see more details in Figure 6 in Appendix A.2). Nevertheless, the accuracy and weighted F1 score of CORECT are 2.89% and 2.75% higher than all baseline models on average. Likewise, we observe the similar phenomena on the IEMOCAP (4-way) dataset with a 2.49% improvement over the previous state-of-the-art models as Table 3. These results affirm the efficiency of CORECT for the multimodal ERC task.

| Methods | Sentiment Classification Accuracy (%) | | Emotion Classification (Binary, 1 vs. all) weighted F1-score (%) | | | | | |
|---|---|---|---|---|---|---|---|---|
| | 2 Class | 7 Class | Happiness | Sadness | Angry | Fear | Disgust | Surprise |
| Multilouge-Net (Shenoy and Sardana, 2020) | 82.88 | 44.83 | 67.84 | 65.34 | 67.03 | *87.79* | 74.91 | *86.05* |
| TBJE (Delbrouck et al., 2020) | 82.40 | 43.91 | 65.91 | 70.78 | 70.86 | *87.79* | 82.57 | 86.04 |
| COGMEN (Joshi et al., 2022) | 82.95 | 45.22 | 70.88 | 70.91 | 74.20 | *87.79* | 81.83 | *86.05* |
| **CORECT (Ours)** | **83.66** | **46.31** | **71.35** | **72.86** | **76.77** | **87.90** | **84.26** | **86.48** |

Table 4: Results on CMU-MOSEI dataset compared with previous works. The **bolded** results indicate the best performance, while the underlined results represent the second best performance.

| Sub-Modules | IEMOCAP (6-way) | | IEMOCAP (4-way) | |
|---|---|---|---|---|
| | Acc. (%) | w-F1 (%) | Acc. (%) | w-F1 (%) |
| -w/o RT-GCN | 66.61 | 66.55 ($\downarrow$ 3.47) | 80.69 | 80.54 ($\downarrow$ 4.10) |
| -w/o P-CM | 66.54 | 66.64 ($\downarrow$ 3.38) | 82.18 | 82.16 ($\downarrow$ 2.48) |
| -w/o $\mathcal{R}_{multi}$ | 66.54 | 66.82 ($\downarrow$ 3.20) | 82.61 | 82.53 ($\downarrow$ 2.11) |
| -w/o $\mathcal{R}_{temp}$ | 67.04 | 67.34 ($\downarrow$ 2.68) | 82.08 | 82.07 ($\downarrow$ 2.57) |
| **CORECT** | **69.93** | **70.02** | **84.73** | **84.64** |

Table 5: The performance of CORECT in different strategies under the fully multimodal (A+V+T) setting. **Bolded** results represent the best performance, while underlined results depict the second best. The $\downarrow$ represents the decrease in performance when a specific module is ablated compared to our CORECT model.

**CMU-MOSEI:** Table 4 presents a comparison of the CORECT model on the CMU-MOSEI dataset with current SOTA models in two settings: Sentiment Classification (2-class and 7-class) and Emotion Classification. Apparently, CORECT consistently outperforms other models with sustainable improvements. One notable observation is the *italicized* results for the *Fear* and *Surprise* labels, where all the baselines have the same performance of 87.79 and 86.05 respectively. During the experimental process, when reproducing these baseline's results, we found that the binary classifiers were unable to distinguish any samples for the *Fear* and *Surprise* labels. However, with the help of technical components, i.e., *RT-GCN* and *P-CM*, our model shows significant improvement even in the presence of severe label imbalance in the dataset. Due to space limitations in the paper, we provide additional experiments on the CMU-MOSEI dataset for all possible combinations of modalities in Table 7 (Appendix A.2).

### 4.3 Ablation study

**Effect of Main Components.** The impact of main components in our CORECT model is presented via Table 5. The model performance on the 6-way IEMOCAP dataset is remarkably degraded when the *RT-GCN* or *P-CM* module is not adopted with the decrease by 3.47% and 3.38% respectively. Similar phenomena is observed on

the 4-way IEMOCAP dataset. Therefore, we can deduce that the effect of *RT-GCN* in the CORECT model is more significant than that of *P-CM*.

For different relation types, ablating either $\mathcal{R}_{multi}$ or $\mathcal{R}_{temp}$ results in a significant decrease in the performance. However, the number of labels may affect on the multimodal graph construction, thus it is no easy to distinguish the importance of $\mathcal{R}_{multi}$ and $\mathcal{R}_{temp}$ for the multimodal ERC task.

Table 8 (Appendix A.2) presents the ablation results for uni- and bi-modal combinations. In the unimodal settings, specifically for each individual modality (A, V, T), it's important to highlight that both *P-CM* module and multimodal relations $\mathcal{R}_{multi}$ are non-existent. However, in bimodal combinations, the advantage of leveraging cross-modality information between audio and text (A+T) stands out, with a significant performance boost of over 2.75% compared to text and visual (T+V) modalities and a substantial 14.54% compared to visual and audio (V+A) modalities.

Additionally, our experiments have shown a slight drop in overall model performance (e.g., 68.32% in IEMOCAP 6-way, drop of 1.70%) when excluding Speaker Embedding $\mathcal{S}_{emb}$ from CORECT.

**Effect of the Past and Future Utterance Nodes.** We conduct an analysis to investigate the influence of past nodes ($\mathcal{P}$) and future nodes ($\mathcal{F}$) on the model's performance. Unlike previous studies

| Modality Settings | IEMOCAP (6-way) | | IEMOCAP (4-way) | |
|---|---|---|---|---|
| | Acc. (%) | w-F1 (%) | Acc. (%) | w-F1 (%) |
| A | 52.31 | 51.49 | 67.02 | 65.48 |
| T | 67.22 | 67.26 | 82.82 | 82.65 |
| V | 38.63 | 37.67 | 49.73 | 47.97 |
| A+T | 68.27 | 68.36 | 83.14 | 83.13 |
| T+V | 65.50 | 65.61 | 81.76 | 81.75 |
| V+A | 54.16 | 53.82 | 69.03 | 68.21 |
| **CORECT (A+T+V)** | **69.93** | **70.02** | **84.73** | **84.64** |

Table 6: The performance of CORECT under various modality settings.

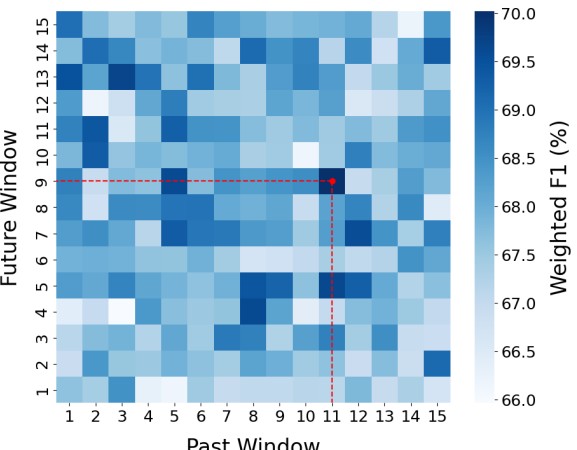

Figure 4: The effects of $\mathcal{P}$ and $\mathcal{F}$ nodes in the past and future of CORECT model on the IEMOCAP (6-way) The red-dash line implies our best setting for $\mathcal{P}$ and $\mathcal{F}$.

(Joshi et al., 2022; Li et al., 2023) that treated $\mathcal{P}$ and $\mathcal{F}$ pairs equally, we explore various combinations of $\mathcal{P}$ and $\mathcal{F}$ settings to determine their effects. Figure 4 indicates that the number of past or future nodes can have different impacts on the performance. From the empirical analysis, the setting $[\mathcal{P}, \mathcal{F}]$ of $[11, 9]$ results in the best performance. This finding shows that the contextual information from the past has a stronger influence on the multimodal ERC task compared to the future context.

**Effect of Modality.** Table 6 presents the performance of the CORECT model in different modality combinations on both the IEMOCAP and CMU-MOSEI datasets.

For IEMOCAP (Table 2 and Table 3), the textual modality performs the best among the unimodal settings, while the visual modality yields the lowest results. This can be attributed to the presence of noise caused by factors, e.g., camera position, environmental conditions. In the bi-modal settings, combining the textual and acoustic modalities achieves the best performance, while combin-

ing the visual and acoustic modalities produces the worst result. A similar trend is observed in the CMU-MOSEI dataset (Table 4), where fusing all modalities together leads to a better result compared to using individual or paired modalities.

# 5 Conclusion

In this work, we propose CORECT, an novel network architecture for multimodal ERC. It consists of two main components including RT-GCN and P-CM. The former helps to learn local-context representations by leveraging modality-level topological relations while the latter supports to infer cross-modal global-context representations from the entire dialogue. Extensive experiments on two popular benchmark datasets, i.e., IEMOCAP and CMU-MOSEI, demonstrate the effectiveness of CORECT, which achieves the new state-of-the-art record for multimodal conversational emotion recognition. Furthermore, we also provide ablation studies to investigate the contribution of various components in CORECT. Interestingly, by analyzing the temporal aspect of conversations, we have validated that capturing the long-term dependencies, e.g., past relation, improves the performance of the multimodal emotion recognition in conversations task.

## Limitations

Hyper-parameter tuning is a vital part of optimizing machine learning models. Not an exception, the learning of CORECT is affected by hyper-parameters such as the number of attention head in P-CM module, the size of Future and Past Window. Due to time constraints and limited computational resources, it was not possible to tune or exploring all possible combinations of these hyper-parameters, which might lead to local-minima convergences. In future, one solution for this limitation is to employ automated hyper-parameter optimization algorithms, to systematically explore the hyperparameter space and improve the robustness of the model. As another solution, we may upgrade CORECT with learning mechanisms to automatically leverage important information, e.g., attention mechanism on future and past utterances.

## Acknowledgements

Cam-Van Thi Nguyen was funded by the Master, PhD Scholarship Programme of Vingroup Innovation Foundation (VINIF), code VINIF.2022.TS143.

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

# A Appendix

## A.1 Implementation Details

### A.1.1 Multimodal Raw Feature Extraction

The multimodal feature extraction process involves extracting features from the acoustic, lexical, and visual modalities for each utterance.

For IEMOCAP, the audio features, with a size of 100, are obtained using the OpenSmile Toolkit (Eyben et al., 2010); visual features, with a size of 512, are extracted using OpenFace (Baltrusaitis et al., 2018); textual features, with a size of 768, are derived using sBERT (Reimers and Gurevych, 2019).

For MOSEI, the audio features are extracted using librosa (McFee et al., 2015) with 80 filter banks, resulting in a feature vector size of 80. The visual features, with a size of 35, are obtained from (Bagher Zadeh et al., 2018). The textual features, with a size of 768, are obtained using sBERT (Reimers and Gurevych, 2019).

### A.1.2 Pairwise Cross-modal Feature Interaction

Figure 5 illustrates details of Pairwise Cross-modal Feature Interaction (P-CM).

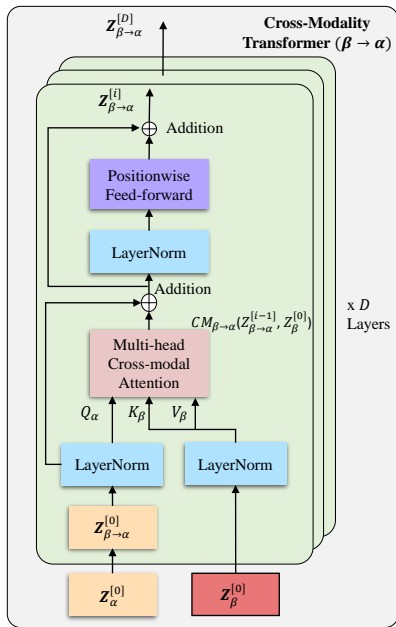

Figure 5: Illustration of the P-CM module between modality $\beta$ and $\alpha$.

## A.2 Additional Experiment Result

Table 8 showcases the results on the IEMOCAP dataset (both 6-way and 4-way) for all the modality combinations of the CORECT model, while Table 7 presents an ablation study conducted on the CMU-MOSEI dataset, considering various modality combination settings.

Figure 6 shows the confusion matrix for prediction on IEMOCAP (4-way) and IEMOCAP (6-way), respectively.

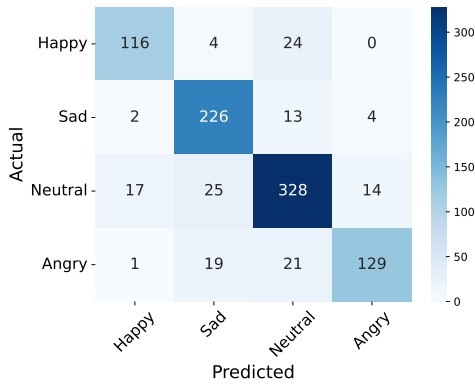

(a) Confusion matrix on the IEMOCAP (4-way).

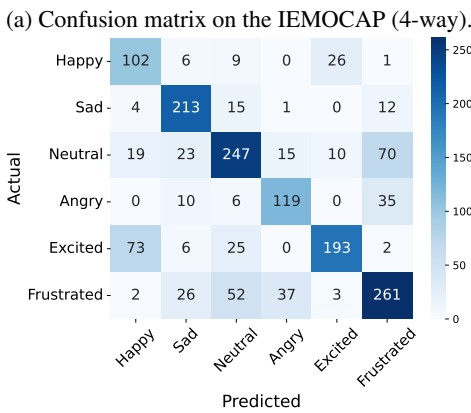

(b) Confusion matrix on the IEMOCAP (6-way).

Figure 6: Visualization the confusion matrices of CORECT under multimodal (A+V+T) setting. Most of False predictions observed on IEMOCAP (6-way) came from the ambiguity between pair of labels: *Happy* and *Excited*, *Neural* and *Frustrate*.

## A.3 Reproducibility

CORECT is implemented using Pytorch [1], and run experiments on Google Colab Pro. We choose Adam as the optimizer and set the dropout rate to 0.5. The numbers of multi-head attentions used in Graph Transformer and P-CM are selected as 7 and 2, respectively. For IEMOCAP dataset, the learning rate is 0.0003; Window size $[\mathcal{P}, \mathcal{F}]$ is tested on various settings in the range of [1,15]. For CMU-MOSEI dataset, the learning rate is 0.0006; Window size $[\mathcal{P}, \mathcal{F}]$ is picked between [5,4] due to the property of short dialogue in CMU-MOSEI. Refer-

---

[1] https://pytorch.org/

| Datasets | Modality Settings | Sentiment Class Accuracy (%) | | Emotion Class weighted F1-score (%) | | | | | |
|---|---|---|---|---|---|---|---|---|---|
| | | 2 Class | 7 Class | Happiness | Sadness | Angry | Fear | Disgust | Surprise |
| Multilogue-Net (Shenoy and Sardana, 2020) | A+T+V | 82.88 | 44.83 | 67.84 | 65.34 | 67.03 | *87.79* | 74.91 | *86.05* |
| TBJE (Delbrouck et al., 2020) | A+T | 82.4 | 43.91 | 65.91 | 70.78 | 70.86 | *87.79* | 82.57 | 86.04 |
| COGMEN (Joshi et al., 2022) | A+T+V | 82.95 | 45.22 | 70.88 | 70.91 | 74.20 | *87.79* | 81.83 | *86.05* |
| **CORECT (Ours)** | T | 84.13 | 45.80 | 67.82 | 72.12 | 75.55 | *87.79* | 84.63 | *86.05* |
| | A+T | **84.28** | 44.89 | 67.49 | 71.53 | 75.39 | *87.79* | **84.69** | *86.05* |
| | A+T+V | 83.66 | **46.31** | **71.35** | **72.86** | **76.77** | **87.90** | 84.26 | **86.48** |

Table 7: Ablation study on CMU-MOSEI dataset. The multimodal implementation (A+V+T) consistently outperformed the baseline models in most of the modality combinations. For the 2-class sentiment and the *Disgust* class emotion, our approach reaches a competitive performance.

| Dataset | | A | | T | | V | | A+T | | T+V | | V+A | | A+V+T | |
|---|---|---|---|---|---|---|---|---|---|---|---|---|---|---|---|
| | | Acc | W-F1 | Acc | W-F1 | Acc | W-F1 | Acc | W-F1 | Acc | F1 | Acc | W-F1 | Acc | F1 |
| IEMOCAP (6-way) | w/o RT-GCN | 35.12 | 30.01 | 64.7 | 64.34 | 30.99 | 26.88 | 67.10 | 66.92 | 65.37 | 65.50 | 52.13 | 51.80 | 66.61 | 66.55 |
| | w/o P-CM | - | - | - | - | - | - | 65.87 | 65.89 | 65.00 | 65.07 | 53.54 | 52.86 | 66.54 | 66.64 |
| | w/o $\mathcal{R}_{multi}$ | - | - | - | - | - | - | 66.30 | 66.27 | 64.76 | 64.78 | 53.67 | 53.48 | 66.54 | 66.82 |
| | w/o $\mathcal{R}_{temp}$ | 41.53 | 39.49 | 63.65 | 63.72 | 27.66 | 27.37 | 67.34 | 67.33 | 65.43 | 65.29 | 50.65 | 49.67 | 67.04 | 67.34 |
| | **CORECT** | **52.31** | **51.49** | **67.22** | **67.26** | **38.63** | **37.67** | **68.27** | **68.36** | **65.50** | **65.61** | **54.16** | **53.82** | **69.93** | **70.02** |
| IEMOCAP (4-way) | w/o RT-GCN | 55.25 | 52.18 | 80.38 | 80.25 | 34.04 | 31.33 | 81.87 | 81.18 | 80.17 | 80.04 | 58.96 | 58.57 | 80.69 | 80.54 |
| | w/o P-CM | - | - | - | - | - | - | 80.91 | 80.94 | 80.38 | 80.04 | 69.25 | 69.00 | 82.18 | 82.16 |
| | w/o $\mathcal{R}_{multi}$ | - | - | - | - | - | - | 81.76 | 81.78 | 80.38 | 80.47 | 69.14 | 68.84 | 82.61 | 82.53 |
| | w/o $\mathcal{R}_{temp}$ | 56.84 | 54.88 | 80.70 | 80.70 | 41.04 | 39.75 | 82.08 | 81.99 | 81.34 | 81.36 | 57.16 | 56.62 | 82.08 | 82.07 |
| | **CORECT** | **67.02** | **65.48** | **82.82** | **82.85** | **49.73** | **47.97** | **83.14** | **83.13** | **81.76** | **81.75** | **69.03** | **68.21** | **84.73** | **84.64** |

Table 8: Ablation study on IEMOCAP dataset. It shows the consistency of our proposal method since any ablation experiments (on both modal settings and modules) results in a reduction of overall performance. On unimodal setting of $\{A, V, T\}$, *P-CM* module and multimodal relation are not exist. Therefore there are no ablation of *P-CM* and $\mathcal{R}_{multi}$ on these unimodal setting, denotes by "-".

ring to the training log on the IEMOCAP (6-way) dataset using Google Colab Pro, each mini-batch (size of 10 dialouges) takes approximately 0.4s. The similar ratio is observed on the MOSEI dataset.