# OpenReview forum: "Conversation Understanding using Relational Temporal Graph Neural Networks with Auxiliary Cross-Modality Interaction"
_EMNLP/2023/Conference — EMNLP 2023 Main_

### Official Review · Reviewer_Xeu3 · 2023-08-04

**Soundness:** 3

**Excitement:**

3: Ambivalent: It has merits (e.g., it reports state-of-the-art results, the idea is nice), but there are key weaknesses (e.g., it describes incremental work), and it can significantly benefit from another round of revision. However, I won't object to accepting it if my co-reviewers champion it.

**Paper Topic And Main Contributions:**

This paper aims to propose the Relational Temporal Graph Neural Network with Auxiliary Cross-Modality Interaction (CORECT), a novel neural network framework that effectively captures conversation-level cross-modality interactions and utterance-level temporal dependencies in a modality-specific manner for conversation understanding.

**Questions For The Authors:**

Question A: Why not compare with some advanced baselines, and what criteria were used to select these baselines?

**I lean towards giving a rating of 2.5 on Excitement.** The performance of the method doesn't seem significantly superior, as observed in various other approaches listed below. Additionally, the paper's contributions appear rather limited, primarily focusing on refining the COGMEN model.

*Shi, T., & Huang, S. L. (2023). MultiEMO: An Attention-Based Correlation-Aware Multimodal Fusion Framework for Emotion Recognition in Conversations. In Proceedings of the 61st Annual Meeting of the Association for Computational Linguistics (Volume 1: Long Papers) (pp. 14752-14766).*
*Chen, F., Shao, J., Zhu, S., & Shen, H. T. (2023). Multivariate, Multi-Frequency and Multimodal: Rethinking Graph Neural Networks for Emotion Recognition in Conversation. In Proceedings of the IEEE/CVF Conference on Computer Vision and Pattern Recognition (pp. 10761-10770).*
*Lian, Z., Liu, B., & Tao, J. (2022). Smin: Semi-supervised multi-modal interaction network for conversational emotion recognition. IEEE Transactions on Affective Computing.*

**Reasons To Accept:**

-	Reasonable method.

**Reasons To Reject:**

-	So difficult to follow the contribution of this paper. And it looks like an incremental engineering paper. The proposed method has been introduced in many papers, such as [1] Joshi, A., Bhat, A., Jain, A., Singh, A., & Modi, A. (2022, July). COGMEN: COntextualized GNN-based Multimodal Emotion Recognition. In Proceedings of the 2022 Conference of the North American Chapter of the Association for Computational Linguistics: Human Language Technologies (pp. 4148-4164).
-	The related work should be updated with more recent related works.
-	The experimental section needs some significance tests to further verify the effectiveness of the method put forward in the paper.
-	For the first time appearing in the text, the full name must be written, and abbreviations must be written in parentheses. When it appears in the abstract, it needs to be written once, and when it appears in the main text, it needs to be repeated again, that is, the full name+parentheses (abbreviations) should appear again.
-	Error analysis plays a crucial role in evaluating model performance and identifying potential issues. We encourage the authors to conduct error analysis in the paper and provide detailed explanations of the model's performance under different scenarios. Error analysis will aid in guiding subsequent improvements and expansions of the ERC research.
-	Writing mistakes are common across the overall paper, which could be found in “Typos, Grammar, Style, and Presentation Improvements”.

**Reproducibility:**

3: Could reproduce the results with some difficulty. The settings of parameters are underspecified or subjectively determined; the training/evaluation data are not widely available.

**Reviewer Confidence:**

3: Pretty sure, but there's a chance I missed something. Although I have a good feel for this area in general, I did not carefully check the paper's details, e.g., the math, experimental design, or novelty.

**Typos Grammar Style And Presentation Improvements:**

Typos, Grammar, Style, and Presentation Improvements:

-	The structure and fluency of the sentences in the paper are poor and require improvement to enhance the quality of the writing. (e.g., “Our social interactions and relationships are all influenced by emotions.”, “MMGCN adopts an undirected graph to effectively fuse multimodal information and capture long-distance contextual and inter-modal interactions”, “Extensive experiments on two popular benchmark datasets, i.e., IEMOCAP and CMU-MOSEI, demonstrate the effectiveness of CORECT”, “we validate that capturing the long-term dependencies, e.g., past relation, improves the performance of multimodal ERC task.”)
-	On page 2, there are some words or punctuation that may have ambiguity. (e.g., “ CORECT is a multimodal ERC model based on GNNs which captures both temporal dependencies and cross-modality interaction in conversational.”) In particular, the above problems appear in many parts of the paper.

---

> ### Author Rebuttal · Authors · 2023-08-28
>
> We would like to thank the reviewer for your time in providing insightful feedback on our work. Your remarks are valuable guidance for us to improve the paper quality. For your questions, we would like to respond as follows:
>
> ### I. Questions:
>
> **Q1.**  Why not compare with some advanced baselines, and what criteria were used to select these baselines. The related work should be updated with more recent related works.
>
> **Answer:** Thank you for your comment. Via a literature review on related works until the submission time (Section 1, Section 2.1), we selected the most prominent baseline models (Section 4.1) for the multimodal Emotion Recognition (ERC) task, and COGMEN [1] is the state-of-the-art one.  However, we will double-check and update recent related works accordingly.
>
> **Q2.** So difficult to follow the contribution of this paper. And it looks like an incremental engineering paper. The proposed method has been introduced in many papers, such as [1] Joshi, A., Bhat, A., Jain, A., Singh, A., & Modi, A. (2022, July). COGMEN: COntextualized GNN-based Multimodal Emotion Recognition. In Proceedings of the 2022 Conference of the North American Chapter of the Association for Computational Linguistics: Human Language Technologies (pp. 4148-4164)
>
> **Answer:** Thank you for sharing your thoughts on the paper. We will work diligently to improve the presentation of our paper's contribution and originality to make it more comprehensible to readers.  Specifically, our approach differs from the COGMEN model in a significant way as follows:
>
> * While COGMEN utilizes fused features from all three modalities as input, we focus on modality-specific features in our approach. The use of fused features might blur the distinct roles of individual modalities within the model. Concatenating input representations is not ideal due to varying representation size of each modality; and the need to independently consider each modality's contributions, rather than pooling them together. Thus, our adoption of modality-specific features addresses the limitations inherent in the COGMEN model.
>
> * Additionally, it's worth noting that COGMEN primarily focuses on harnessing contextual information of speaker flow emotions through inter-speaker and intra-speaker relations within a conversation. However, we believe that this approach does not fully exploit the rich information present in the conversation. Our model, on the other hand, operates at the utterance-level, where each utterance encapsulates abundant information including context, speaker identity, and emotions. We take into account the intricate interplay of these factors, including their past and future relationships, to achieve a comprehensive understanding of the conversational data.
>
> ### II. Suggestions:
>
> **S1.** The experimental section needs some significance tests to further verify the effectiveness of the method put forward in the paper.
>
> **Answer:** Thank you for your comment. We follow the evaluation strategy of COGMEN [1], in which only the best performance is reported. Nevertheless, we will conduct significance tests and update the result in our final version.
>
> **S2.** Error analysis plays a crucial role in evaluating model performance and identifying potential issues. We encourage the authors to conduct error analysis in the paper and provide detailed explanations of the model's performance under different scenarios. Error analysis will aid in guiding subsequent improvements and expansions of the ERC research.
>
> **Answer:** Thank you for your suggestion. Due to the space limitation, we briefly discuss potential issues for each experimental result in Section 4.2 and Section 4.3. We will consider to add the error analysis subsection to our final version.
>
> ### III. Typos, Grammar, Style, and Presentation Improvements:
> The structure and fluency of the sentences in the paper are poor and require improvement to enhance the quality of the writing ... On page 2, there are some words or punctuation that may have ambiguity
>
> **Answer:** Thank you for pointing out the issues. We will carefully go through your feedback and fix all the errors in the final version.

---

### Official Review · Reviewer_YYyo · 2023-08-05

**Soundness:** 4

**Excitement:**

4: Strong: This paper deepens the understanding of some phenomenon or lowers the barriers to an existing research direction.

**Paper Topic And Main Contributions:**

The paper focuses on the task of Emotion Recognition in Conversations (ERC), particularly in a multimodal context, where emotions are recognized from text content, facial expressions, and audio signals in a conversation.

The main contributions of the paper lie in proposing the CORECT framework, which effectively captures temporal dependencies and cross-modal interactions in multimodal ERC tasks, and demonstrates its superiority over existing approaches through comprehensive experiments on real-world datasets. The research addresses important challenges in understanding emotions in conversations, particularly in the context of diverse modalities, and contributes to advancements in multimodal emotion recognition.

**Questions For The Authors:**

Same as the Reasons To Reject.

**Reasons To Accept:**

1. The paper identifies and addresses the limitations of existing methods for multimodal ERC. It introduces the CORECT framework, which effectively captures both temporal dependencies and cross-modal interactions in conversations, thereby enhancing the overall performance of emotion recognition.

2. The paper conducts extensive experiments on real-life datasets (IEMOCAP and CMU-MOSEI) to evaluate the performance of CORECT. The results demonstrate that the proposed model consistently outperforms state-of-the-art baselines, highlighting its effectiveness.

3. The paper is well-structured and presents the proposed framework and experimental results in a clear and concise manner. The authors effectively convey the motivation, methodology, and results, making it easy for readers to understand the contributions and implications of the research.

**Reasons To Reject:**

1. The evaluation is conducted on publicly available datasets (IEMOCAP and CMU-MOSEI), which have been extensively used in previous research. While these datasets are valuable resources, using them alone may not be sufficient to fully validate the proposed model's effectiveness and generalization to other domains or real-world scenarios.

2. The paper briefly mentions the limitations of previous works as motivation for proposing CORECT. However, it does not adequately discuss the limitations of the proposed model itself. And the limitation section seems to be added for a short time before the submission deadline.

**Reproducibility:**

4: Could mostly reproduce the results, but there may be some variation because of sample variance or minor variations in their interpretation of the protocol or method.

**Reviewer Confidence:**

4: Quite sure. I tried to check the important points carefully. It's unlikely, though conceivable, that I missed something that should affect my ratings.

---

> ### Author Rebuttal · Authors · 2023-08-28
>
> We extend our sincere appreciation to the reviewer for dedicating time to offer perceptive and precious feedback on our work. Your comments are beneficial in directing us to enhance the quality of our research. We would like to provide responses for your concerns as follows:
> ### I. Questions:
> **Q1.** The evaluation is conducted on publicly available datasets (IEMOCAP and CMU-MOSEI), which have been extensively used in previous research. While these datasets are valuable resources, using them alone may not be sufficient to fully validate the proposed model's effectiveness and generalization to other domains or real-world scenarios.
>
> **Answer:** Thank you for highlighting this concern. As mentioned in Section 4.1, IEMOCAP and CMU-MOSEI are popular benchmark datasets for the ERC task. We intend to further investigate the effectiveness of CORECT in other contexts and domains as part of our future research.
>
> **Q2.** The paper briefly mentions the limitations of previous works as motivation for proposing CORECT.  However, it does not adequately discuss the limitations of the proposed model itself. And the limitation section seems to be added for a short time before the submission deadline.
>
> **Answer:** Thank you for your comment. We will improve the discussion on the limitation of CORECT in our final version.

---

### Official Review · Reviewer_bUo5 · 2023-08-07

**Soundness:** 4

**Excitement:**

4: Strong: This paper deepens the understanding of some phenomenon or lowers the barriers to an existing research direction.

**Missing References:**

The literature review is sufficiently systematic and comprehensive.

**Paper Topic And Main Contributions:**

The authors design a novel framework, called CORECT, for addressing Multimodal Emotion Recognition in Conversations (multimodal ERC). CORECT consists of two core components, namely RT-GCN and P-CM. RT-GCN is devised to capture utterance-level local context features, whereas P-CM is proposed for extracting cross-modal global context features at the conversation level. These two kinds of features are then fused to boost the performance of the utterance-level emotional recognition.

**Questions For The Authors:**

1. In the unimodal encoder, why is the text modality implemented with a transformer, while the other two modalities use FC (fully connected layers)? What is the underlying rationale for this inconsistency?

2. Is the speaker embedding represented as a one-hot encoding? If not, how is it generated? Additionally, is it a necessary component? Is there a significant performance drop if the speaker embedding is not used?

3. In Table 5, why does the combination of text and visual modalities (T+V) perform worse than using the text modality (T) alone?


**Reasons To Accept:**

1. The motivation is clear, and the proposed CORECT seems to be a reasonable solution.
2. The experiments are extensive, comparing with various previous works. The ablation study sufficiently demonstrates the effectiveness of RT-GCN and P-CM, and the article also examines the impact under different modalities.
3. Well-written and easy to follow. The literature review is sufficiently systematic and comprehensive.

**Reasons To Reject:**

Lack of complexity analysis for the model. Overall, CORECT is a rather complex framework, incorporating various neural network components such as Transformer and GNN. It is essential for the authors to assess the computational efficiency of this framework and compare it with the baseline to demonstrate the practical applicability of this method in real-world scenarios.

**Reproducibility:**

4: Could mostly reproduce the results, but there may be some variation because of sample variance or minor variations in their interpretation of the protocol or method.

**Reviewer Confidence:**

4: Quite sure. I tried to check the important points carefully. It's unlikely, though conceivable, that I missed something that should affect my ratings.

**Typos Grammar Style And Presentation Improvements:**

Typos:
Line 377, "difficulty of of" should be "difficulty of";
Line 388, "are" should be removed;
Line 391, "Querys" should be "Queries"
Presentation Improvements:
1. In the introduction, it would be helpful to include a figure to provide a more visual representation of the article's motivation. Additionally, it is recommended to include some preliminary validation experiments to support the author's argument that modality-specific representations are superior to fused representation.

2. Consider incorporating some case studies and visualizations of intermediate results in P-CM to aid readers in better understanding why the proposed method can enhance performance.

---

> ### Author Rebuttal · Authors · 2023-08-28
>
> We would like to express our heartfelt gratitude to the reviewer for taking the time to provide such insightful and valuable comments on our work. Your feedback has been incredibly helpful in guiding us towards improvements for our shortcomings. We hope that our responses would better clarify what we do and contribute to the research problem. We would like to address your concerns with the following responses:
>
> ### I. Questions:
>
> **Q1.** In the unimodal encoder, why is the text modality implemented with a transformer, while the other two modalities use FC (fully connected layers)? What is the underlying rationale for this inconsistency?
>
> **Answer**: Thank you for your interesting question. We adopt similar multimodal encoders as our state-of-the-art baseline, e.g., COGMEN [1], which uses the transformer-based encoder to capture the semantic context of each utterance in the textual modality and fully connected (FC) components for other modalities. The encoding strategy shows an efficacy for the multimodal emotion recognition (ERC) task.
>
> **Q2.** Is the speaker embedding represented as a one-hot encoding? If not, how is it generated? Additionally, is it a necessary component? Is there a significant performance drop if the speaker embedding is not used?
>
> **Answer**: Yes, the speaker embedding is a one-hot encoding of the speaker's identity within the conversation. Our experiments have shown a slight drop in overall model performance (e.g., 68.32% in IEMOCAP 6-way, drop of 1.70%) when excluding this embedding. Due to the space limitation, we have not shown this result in the current version. We will update it in the final version.
>
> **Q3.** In Table 5, why does the combination of text and visual modalities (T+V) perform worse than using the text modality (T) alone?
>
> **Answer**: As we mentioned in Section 4.3 about “Effects of Modality”, the reason could be that the visual modality contains noise, e.g., camera position or environmental conditions, which might negatively impact overall performance. Another factor could be the challenge of effectively integrating both modalities due to the complexity of aligning and fusing different types of data.
>
>
> ### II. Suggestions:
>
> **S1.** Lack of complexity analysis for the model ... It is essential for the authors to assess the computational efficiency of this framework and compare it with the baseline to demonstrate the practical applicability of this method in real-world scenarios."
>
> **Answer**: Thank you for suggesting us to demonstrate computational efficiency of our framework CORECT. Referring to the trainining log on the IEMOCAP (6-way) dataset using Google Colab Pro, each mini-batch (size of 10 dialouges) takes approximately 0.3s and 0.4s for COGMEN [1] and CORECT respectively. The similar ratio is observed on the MOSEI dataset. We will update this comparison in the Reproducibility section of our final version.
>
> **S2**. In the introduction, it would be helpful to include a figure to provide a more visual representation of the article's motivation
>
> **Answer**: Thank you for your suggestion, we will update it in the final version.
>
> **S3**. Additionally, it is recommended to include some preliminary validation experiments to support the author's argument that modality-specific representations are superior to fused representation
>
> **Answer**: As demonstrated in Section 4.2, our model CORECT consistently outperform baseline models, including COGMEN [1] which relies on fused features as input. In our model, if fused features are used as input, P-CM module will be disable. The impact of removing the P-CM module has been explored and discussed in Section 4.3 within the "Effect of Main Components" subsection. The results demonstrate that the absence of the P-CM module indeed affects the overall model performance, leading to a reduction of 3.38% in IEMOCAP (6-way) and 2.48% in IEMOCAP (4-way).
>
> **S4**. Consider incorporating some case studies and visualizations of intermediate results in P-CM to aid readers in better understanding why the proposed method can enhance performance.
>
> **Answer**: Thank you for your great suggestion. We already run an ablation analysis on the P-CM module and consistently observe that P-CM outperforms its ablation counterparts. These findings will be included in the final version of our paper.
>
> ### III. Typos Grammar Style And Presentation Improvements:
> Typos: Line 377, "difficulty of of" should be "difficulty of"; Line 388, "are" should be removed; Line 391, "Querys" should be "Queries" Presentation Improvements:
>
> **Answer**: Thank you for pointing out these typos. We will fix them in our final version.
>
> ### IV. References:
>
> [1] Joshi, A., Bhat, A., Jain, A., Singh, A., & Modi, A. (2022, July). COGMEN: COntextualized GNN-based Multimodal Emotion Recognition. In Proceedings of the 2022 Conference of the North American Chapter of the Association for Computational Linguistics: Human Language Technologies (pp. 4148-4164)

---

### Meta-Review · Area_Chair_XDau · 2023-09-17

**Recommendation:** 4

**Metareview:**

This paper introduces a novel multimodal framework, CORECT for emotion recognition in conversations. The proposed architecture consists of two major component to capture the local context features in utterance-level along with the cross-modal global context features in conversation-level. The authors shows some modest gain in performance of CORECT when compared with other multimodal ERC research using two widely used datasets.

All the reviewers agree that the current study provides sufficient support for its claim and offers meaningful advancements over existing emotion recognition methods. However, the reviewers have identified some minor issues, such as:
1. An insufficient discussion regarding the limitations of the proposed framework.
2. Lack of complexity analysis, more focusing on computational efficiency of the model for practical implementation among others.

The authors have addressed some of these concerns in the comment and in the rebuttal.  By addressing the reviewer’s comments, the paper quality will improve substantially.
I believe this paper will be interesting to the Speech community.

---

### Decision · Program_Chairs · 2023-10-07

**Decision:**

Accept-Main

**Comment:**

This paper introduces a novel multimodal framework, CORECT for emotion recognition in conversations. The proposed architecture consists of two major component to capture the local context features in utterance-level along with the cross-modal global context features in conversation-level. The authors shows some modest gain in performance of CORECT when compared with other multimodal ERC research using two widely used datasets.

All the reviewers agree that the current study provides sufficient support for its claim and offers meaningful advancements over existing emotion recognition methods. However, the reviewers have identified some minor issues, such as:
1. An insufficient discussion regarding the limitations of the proposed framework.
2. Lack of complexity analysis, more focusing on computational efficiency of the model for practical implementation among others.

The authors have addressed some of these concerns in the comment and in the rebuttal.  By addressing the reviewer’s comments, the paper quality will improve substantially.
I believe this paper will be interesting to the Speech community.